# Autonomous nanorobots with powerful thrust under dry solid-contact conditions by photothermal shock

Zhaoqi Gu[1,4], Runlin Zhu [1,4], Tianci Shen[1,4], Lin Dou[1], Hongjiang Liu[1], Yifei Liu[1], Xu Liu[2], Jia Liu[3], Songlin Zhuang[1] & Fuxing Gu [1] ✉

Nanorobotic motion on solid substrates is greatly hindered by strong nano-friction, and powerful nanomotors–the core components for nanorobotic motion–are still lacking. Optical actuation addresses power and motion control issues simultaneously, while conventional technologies with small thrust usually apply to fluid environments. Here, we demonstrate micronewton-thrust nanomotors that enable the autonomous nanorobots working like conventional robots with precise motion control on dry surfaces by a photothermal-shock technique. We build a pulsed laser-based actuation and trapping platform, termed photothermal-shock tweezers, for general motion control of metallic nanomaterials and assembled nanorobots with nanoscale precision. The thrust-to-weight ratios up to $10^7$ enable nanomotors output forces to interact with external micro/nano-objects. Leveraging machine vision and deep learning technologies, we assemble the nanomotors into autonomous nanorobots with complex structures, and demonstrate multi-degree-of-freedom motion and sophisticated functions. Our photothermal shock-actuation concept fundamentally addresses the nanotribology challenges and expands the nanorobotic horizon from fluids to dry solid surfaces.

Robots, the machines that sense, think and act, bring lots of convenience to humanity, while nanorobots represent revolutionary technologies in the future[1]. Powerful nanomotors is essential for nanorobots to propel themselves, move components or manipulate external objects[2–7]. At nanoscale, the interaction between two objects in contact is governed by interfacial adsorption and friction, which are induced by van der Waals forces and much stronger than gravity and inertial forces. Conventional nanomotor technologies utilize thrusts from the momentum of particles in the ambient medium, such as photophoretic force[8], thermophoretic force[9], optothermal trapping force[10] and chemical fuel (catalysis) propulsion force[11], and convection-induced force, and they usually produce small thrust (-$10^{-12}$ N). Overcoming nanofriction (-$10^{-6}$ N) on dry substrates or component contact

is a critical challenge[12–14], which cannot be directly realized by these technologies, and thus previous researches mainly investigate nanomotors or nanorobots in fluid environments[8–11]. Meanwhile, the thrust generated by widely used optical tweezers, originating from momentum transfer of photons, is also too small (-$10^{-12}$ N) to overcome the interface friction[15] (Supplementary Note 1), so their applications are also limited to the fluid environments. The lack of powerful thrust from nanomotors–the core components of nanorobots for locomotion[16,17], is inhibiting various potential applications of nanorobots on dry solid-contact conditions where space or mass is limited, such as endoscopies, space operations, and microdrone missions[5]. Despite advances in reducing nanofriction through surface lubrication (e.g., solid-liquid phase transition) and contact minimization

[1]Laboratory of Integrated Opto-Mechanics and Electronics, Shanghai Key Laboratory of Modern Optical System, School of Optical-Electrical and Computer Engineering, University of Shanghai for Science and Technology, 200093 Shanghai, China. [2]State Key Laboratory of Reliability and Intelligence of Electrical Equipment, Hebei University of Technology, 300130 Tianjin, China. [3]Department of Industrial and Systems Engineering, Auburn University, Auburn, AL 36849, USA. [4]These authors contributed equally: Zhaoqi Gu, Runlin Zhu, Tianci Shen. ✉e-mail: gufuxing@usst.edu.cn

(e.g., curved surfaces)[13,18–22], the applicable actuation conditions are relatively limited due to the requirement of specific conditions and complex structures, and none further demonstrate motion control and outputting forces against external loading, which are essential capabilities for robots. Fundamentally, a general actuation technology endowing nanomotors with sufficient thrust can solve the nanofriction barriers, which is urgently needed for practical applications but remains unsolved.

According to Newton's second law of motion and impulse-momentum theorem, large forces ($F$, Fig. 1a) or accelerations ($a$) can be induced by short excitations ($\Delta t$), such as collisions, shocks or explosions. In particular, under excitation by high-intensity pulsed lasers, transient thermal expansion will exert an impulsive load inside objects, i.e., photothermal shock. Objects are often driven by exciting their inherent vibration modes[21,23–26]. In contrast, due to the transient and high-energy nature of laser pulses, the initial impulsive excitation of the shock load deposits sufficient energy inside the object to produce forces or accelerations that are considerably greater than those generated by the later-established stable vibration modes, similar to the large acceleration of snakes when striking relative to their slow crawling locomotion (Fig. 1a, inset)[27]. Furthermore, through local or nonuniform excitation of nano-objects, asymmetric spatiotemporal distributions of forces or accelerations may break the symmetry in frictional contacts and generate net displacements.

In this work, we demonstrate a simple and general actuation technique that endows nanomotors with powerful thrust, based on this photothermal-shock concept. We use conventional metallic nanomaterials including nanowires and nanoplates as nanomotor prototypes, due to the high photothermal conversion capabilities[28,29]. We build a pulsed laser-based actuation platform, termed photothermal-shock tweezers ("Methods" and Supplementary Fig. 1), which can trap micro/nano-objects in light spots that like conventional

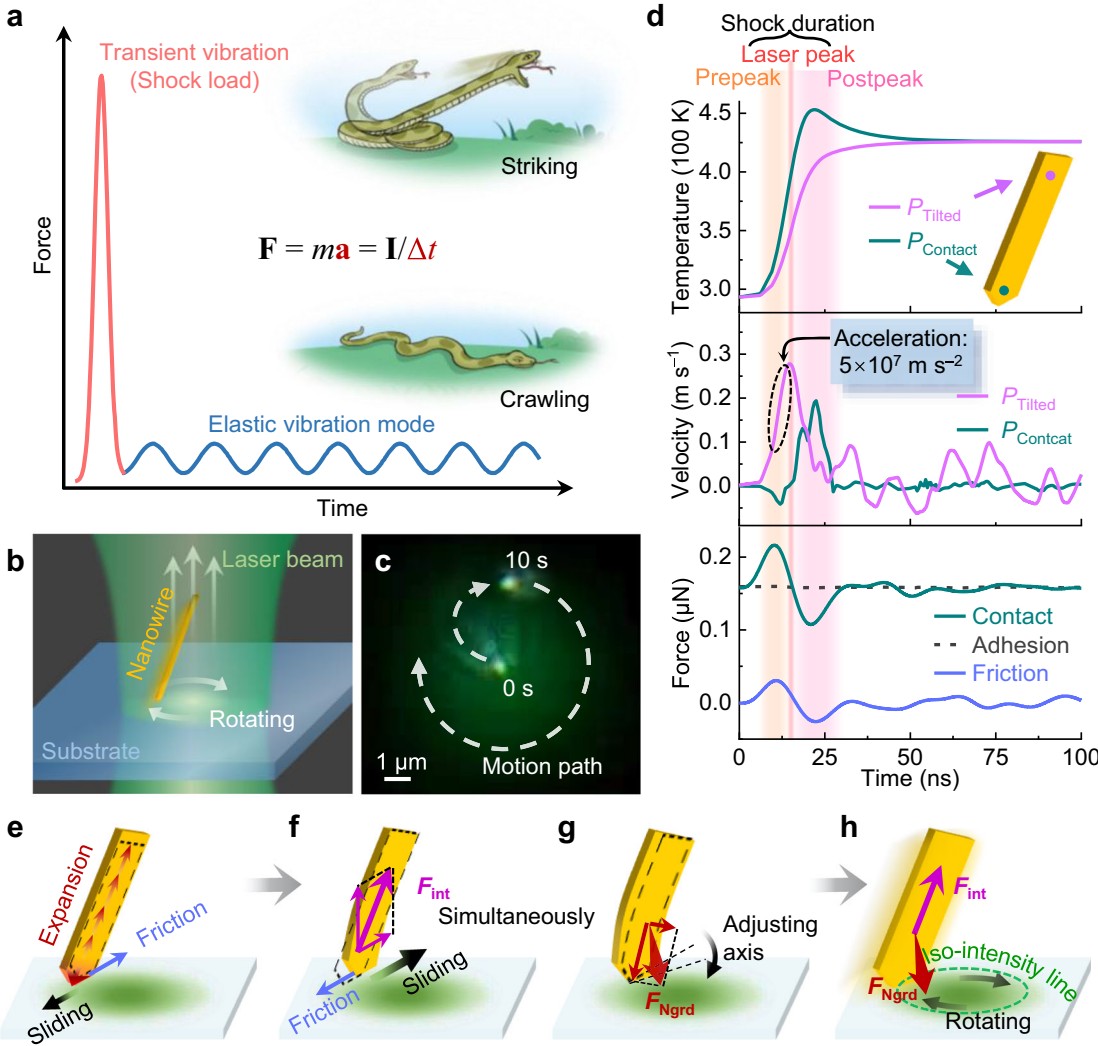

**Fig. 1 | Impulsive forces and accelerations via photothermal shock. a** Schematic comparison between the forces produced by shock-induced transient vibrations and later-established stable elastic vibration modes, which can be roughly described by the two motion types (inset) of a snake quickly striking (top) and slowly crawling (bottom). **b** Schematic depicting a tilted nanowire rotating on a substrate. **c** Sequential superimposed motion track of a 3.7-μm-length nanowire leaving from the spot center and rotating, confirming a trapping behavior. **d** Time-dependent simulation results of temperature, horizontal velocity, and interfacial force. Two typical positions denoted $P_{Contact}$ and $P_{Tilted}$ are located 7 nm and 2.5 μm away from the contact end on the nanowire axis, respectively. The thermal shock duration is divided into two stages, prepeak and postpeak, which are bounded by the laser peak (red line at 15 ns). **e–h** Schematic depicting the motion mechanism. The initial transient thermal expansion (**e**) pushes the contact end slightly towards the lower left, but pushes the tilted end towards the higher right, which freely expands with a large acceleration and produces a strong $F_{int}$ (**f**). Simultaneously, the nanowire axis is adjusted by $F_{Ngrd}$ (**g**). Under the synergistic action of $F_{int}$ and $F_{Ngrd}$, the nanowire rotates along the iso-intensity line (**h**). The unlabeled arrows on either side of $F_{int}$ (**f**) and $F_{Ngrd}$ (**g**) represent their respective components. Source data are provided as a Source data file.

optical tweezers[15,30–32], and thus are capable of precise motion control of micro/nano-objects. Previous reports have only completed actuation but lacked higher-dimensional motion control[13,20,21,25], or completed the motion guidance of collective particles but lacked the precise motion control of individual particles[33]. Our technique features unique trapping behavior, and simultaneously addresses the challenging issues of actuation and motion control, enabling nanomotors locomoting with the guidance of light spots and thus free two-dimensional motion control on dry solid surfaces. Furthermore, our nanomotors can output forces against external loading and be combined with artificial intelligent motion control for autonomous nanorobot applications, such as carrying cargo, moving onboard components, in situ sensing, assembling artificial molecules, and cleaning up nanodroplets.

## Results

### Actuation and motion control

To investigate the motion mechanism under various contact conditions, metallic nanomaterials were randomly dispersed on silica substrates (Supplementary Fig. 2). We first investigated the simplest point-contact condition with a gold nanowire tilted on a silica substrate (Fig. 1b). The dihedral angle at the end[34,35] allowed the nanowire to adhere stably to the substrate with a tilted angle (Supplementary Fig. 3). Under 532-nm-wavelength irradiation with a spot diameter ($D_{spot}$) of 10 μm, a repetition frequency ($R_{rep}$) of 100 Hz and an average power ($P_{ave}$) of 0.5 μW, a 3.7-μm-length nanowire quickly left the light spot center and then moved clockwise in a quasi-circular motion (~6.5 μm in diameter) after reaching an equilibrium position (Fig. 1c and Supplementary Movie 1), confirming a trapping-like behavior. The average velocity of the quasi-circular motion was ~0.5 μm s$^{-1}$, corresponding to a step resolution of ~5 nm per pulse.

The tilted nanowire operated as a rotational nanomotor confined around the spot, but such a trapping-like motion property cannot be explained by the theory of elastic waves[21,25] or light momenta[15,30,32]. Simulations ("Methods" and Supplementary Fig. 3) show that upon initiation of thermal shock (prepeak), the highest temperature occurs at the nanowire contact end (Fig. 1d), rising ~150 K near the pulse peak. The induced thermal expansion towards the lower left is hindered by friction (Fig. 1e), while the expansion pushes the semi-suspended upper part towards the upper right at high accelerations ($10^7$ m s$^{-2}$, Fig. 1d) and strain rates ($10^5$ s$^{-1}$). During the second half of the laser pulse duration (postpeak), the upper part maintains the initiated motion due to inertia (Fig. 1f). Even if the nanowire mass is very small (~$10^{-12}$ g), such a large acceleration produces a strong thermal inertia force ($F_{int}$) that pulls the contact end, similar to thermal inertia motors[36]. The horizontal component of $F_{int}$ can overcome the maximum stiction and pulls the contact end to slide towards the tilted end, with the vertical component decreasing the contact force and thus maximum stiction (Fig. 1f). After the thermal shock, the tilted end continues vibrating (Fig. 1d and Supplementary Movie 2), but the forces induced by established elastic vibration modes are too weak to overcome the friction. The calculated forward net displacement of the nanowire is ~1.2 nm per pulse, which roughly agrees with the experimental result.

Under the action of $F_{int}$, the nanowire will only step straightly in its tilted end direction. There should be another force causing tilted nanowire to rotate around the light spot. Due to the presence of light intensity gradient, the nanosecond pulsed light spot induces the nanowire to generate a transient temperature gradient, which produces an additional thermal gradient force ($F_{grd}$, Fig. 1g), given by

$$\mathbf{F}_{grd} = \mu\alpha\nabla T, \qquad (1)$$

where $\mu$ is Lamé's second coefficient, $\alpha$ is thermal expansion coefficient, and $\nabla T$ is the temperature gradient. We can regard $F_{grd}$ as an impulsive excitation in transient forced vibration, governed by a modified dynamical equation of thermoelasticity

$$\rho\ddot{\mathbf{u}} - \mu\nabla^2\mathbf{u} \approx \mu\alpha\nabla T, \qquad (2)$$

where $\rho$ is the density, $\mathbf{u}$ is the displacement vector and $\ddot{\mathbf{u}}$ denotes the second time derivative of $\mathbf{u}$. When a tilted nanowire is in the light spot center, the thermal inertia force is large, but the thermal gradient force is small, so the nanowires move almost in a straight line and the direction change is small. $F_{grd}$ is actually the force density (per unit volume), and the total force is represented by $F_{Ngrd}$, which is derived from volumetric integration of $F_{grd}$ on the specific region of the nanowire. As shown in Fig. 1g, $F_{Ngrd}$ contains a component towards the more heated contact end and another component towards the light spot center. Therefore, by the principle that $F_{int}$ provides the stepping force and $F_{grd}$ provides the direction control force, the nanowire can adjust its axis perpendicular to the light intensity gradient to make the step and direction adjusting angle of the nanowire under a single pulse meet the conditions of circular motion as it moves (i.e., along the iso-intensity line, Fig. 1h), resulting in a stable quasi-circular motion. Using an orbit-by-orbit method or moving the laser spots directly, the tilted nanowire can locomote anywhere on the substrate (Supplementary Fig. 4 and Supplementary Movie 1).

We then investigated the line-contact condition of nanowires lying on a substrate (Fig. 2a). Upon irradiation ($P_{ave} = 0.9$ μW, $D_{spot} = 8.6$ μm) at one end, a 10.1-μm-length gold nanowire moved into the light spot until their centers roughly coincided (Fig. 2b and Supplementary Movie 3). The average velocity exhibited a slow-fast-slow pattern (Fig. 2c, ~1.3 nm per pulse in the linear range), which is consistent with the gradient distribution of the spot intensity. Under the excitation of higher $R_{rep}$, this process will be significantly accelerated. Unlike the tilted nanowire that always moves towards the cooler end (tilted end direction), the lying nanowire always moves toward the hotter end, indicating that the lying nanowires are not pulled by $F_{int}$. For the lying nanowire, the thermal shock near the hotter end (Fig. 2d and Supplementary Fig. 5) does not have time to affect the colder end via thermal or mechanical conduction. $F_{Ngrd}$ is not counteracted in time by the propagation effect (described by $F_{Npro}$, the volumetric integration of $-\mu\nabla^2\mathbf{u}$) in the nanowire (Supplementary Movie 4). The remaining part (colder end) of the nanowire is governed by static friction and thus remains relatively stationary with the substrate. As a result, the nanowire centroid accelerates ($10^8$ m s$^{-2}$) towards the hotter end due to the net force with a strain rate up to $10^5$ s$^{-1}$. After the thermal shock, the nanowire gradually tends to thermal equilibrium, and in this process the nanowire centroid displacement slightly decreases, but it can be ignored compared with the huge advancing displacement during the thermal shock (Supplementary Fig. 5). Therefore, the net centroid displacement of the nanowire is realized under a single pulse. It is worthy to point out that we can describe the shock driving mechanism either by the volumetric forces ($F_{Npro}$ and $F_{Ngrd}$), or by the boundary condition (friction), so friction indeed plays a crucial auxiliary role in the driving process. The calculated displacement forward is ~3 nm per pulse, which roughly agrees with the experimental result.

The nanowire is covered by two spot regions with opposite light gradients, so it is subject to opposite thermal gradient forces simultaneously. As the nanowire is penetrating into the light spot, the thermal gradient forces opposite the advancing direction is increasing. Therefore, under repeated laser pulses, the nanowire continues locomoting into the light spot until their centers roughly coincide, when the distributions of $F_{grd}$ on the nanowire are balanced (Fig. 2d). As the substrate is slightly moved in the axial direction, the nanowire remains trapped in the light spot center, and thus slide to the center of the light spot, which realizes motion control of the nanowire by guardian of the spot. The lying nanowire acts as a linear nanomotor. In addition, benefiting from polarization-dependent absorption[37], gold nanowires

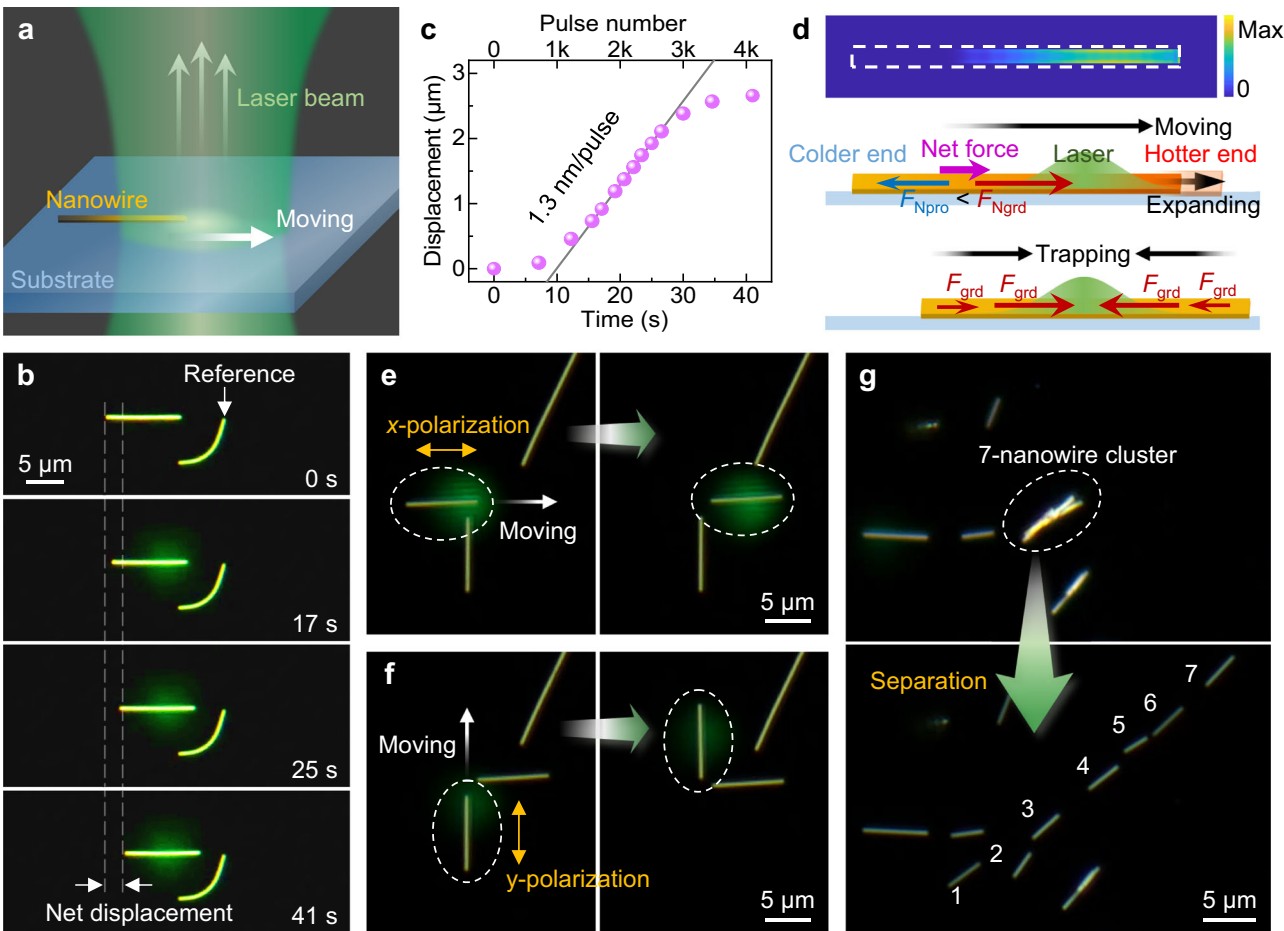

**Fig. 2 | Directional motion control by trap-like behaviors. a** Schematic describing the trap-like behavior of a nanowire lying on a substrate in the light spot. **b** Trap-like process of a 10.1-μm-length nanowire into the spot center. **c** Time-dependent experimental displacements in (**b**), with a linear range of 0.13 μm s⁻¹, corresponding to 1.3 nm per pulse. **d** Schematic describing the basic control mechanism. Top: normalized heat absorption distribution (Max: $5.0 \times 10^{17}$ m⁻³) on the nanowire bottom surface. Middle: the nanowire moves towards the hotter end when subjected to one-side thermal shock, with a net force induced by $F_{Ngrd}$ exceeding $F_{Npro}$. Bottom: under repeated pulses, the nanowire is finally trapped in the spot center, with the $F_{grd}$ values of each part balanced. The green shaded areas represent the Gaussian-like light spot. **e**, **f** Selective control of two orthogonal nanowires (marked in dashed ellipse) by using **e** x- and **f** y-axis polarized lasers (labeled by the yellow bidirectional arrows), respectively. The white arrows represent the moving direction of the nanowire under the guidance of the laser beam. **g** Separation of a 7-nanowire cluster (top) into individual nanowires (bottom), by moving the light spot forward and back along the axis of nanowires. Source data are provided as a Source data file.

can move more easily by light polarized parallel to their axis rather than the perpendicular polarized, which is helpful for selectively controlling nanowires with different orientations (Fig. 2e, f). For nanowires that are stuck together, such as a 7-nanowire cluster depicted in Fig. 2g, each nanowire can easily be separated by simple pulsed lasers irradiation and moving the light spot forward and back along the axis of nanowires (Supplementary Movie 5). Furthermore, the nanowires were observed to bounce off the substrate at relative high intensity of the pulsed laser (Supplementary Movie 6).

As $P_{ave}$ was increased to 1.2 μW, we observed that the nanowire trapped in the spot center bent along the iso-intensity line (Fig. 3a, b and and Supplementary Movie 7), which cannot be explained by elastic waves as well. The thermal gradient forces still exist, and they squeeze the nanowire towards the center at both ends respectively. The nanowire is stable in the axis direction, but unstable in the lateral direction. If the nanowire and the spot are not strictly coincident, or the nanowire itself has certain initial lateral bending, the nanowire will deviate from the equilibrium and rapidly bend, allowing the nanowire to overcome friction laterally and bend along an iso-intensity line (Fig. 3c). As the substrate is slightly moved towards the opposite bending direction, the nanowire remains bending along the iso-intensity line and laterally slides, which realizes the lateral motion

control of the nanowire under repeated pulses (Fig. 3d). We found that the minimum $P_{ave}$ required for actuating nanowire was ~0.3 μW (corresponding to a pulse energy of 3 nJ, Supplementary Fig. 6), which is at least 4 orders of magnitude lower than that used in conventional optical tweezers. Thus, we have realized the basic motion control of nanowires, by which more complex motion can be accomplished such as steering of nanowire axis. For example, we used multiple gold nanowires to assemble the English word *SHOCK* and its corresponding Chinese character *CHONG* (Fig. 3e, f), suggesting the possibility of realizing group motion control of multi-nanowires by multiple laser beams simultaneously.

Our technique based on photothermal shock is general and versatile by changing some configurations, such as target geometries, target materials, actuation wavelengths, spot shapes, and substrate materials. For surface-contact conditions, such as nanoplates on the substrate, although nanofriction is much higher, nanoplates can be trapped and controlled by increasing $D_{spot}$ or increasing $P_{ave}$. Except for gold, we achieved driving for palladium (Pd, Supplementary Fig. 7 and the next section) and silver (Supplementary Movie 8). We also utilized spatial light modulation to realize the manipulation by annular light spots (Supplementary Movie 9). In addition, we used different substrates to drive nanowires, such as magnesium fluoride, the silica

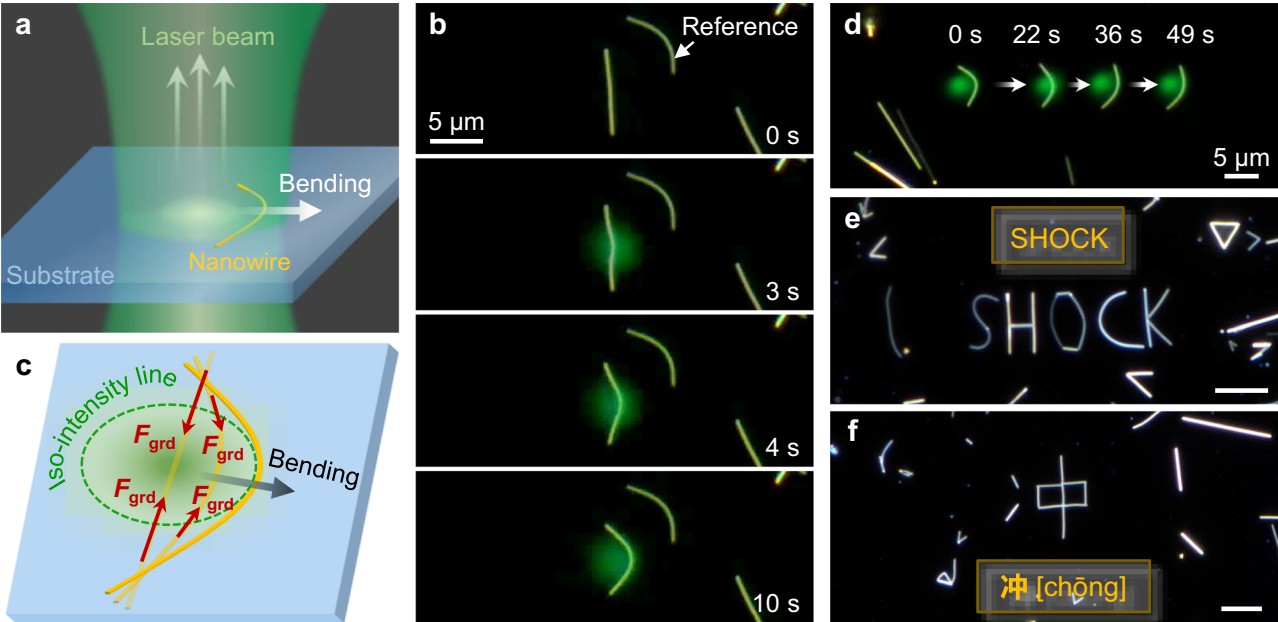

**Fig. 3 | Complex motion control. a** Schematic describing lateral bending of a nanowire in the spot center. **b** Bending process of the nanowire (same as in Fig. 2b) as $P_{ave}$ increased. **c** Schematic describing the lateral motion mechanism. **d** Superimposed photographs of a nanowire at four sequential moments, with the nanowire moving laterally for a long-distance and the background remaining unchanged. **e**, **f** Photographs showing the English word *SHOCK* (**e**) and its corresponding Chinese character *CHONG* (**f**) assembled with multiple nanowires. Scale bar, 5 μm.

substrate covered by monolayer $MoS_2$ and the polystyrene substrate (Supplementary Fig. 9).

**Practical intelligent nanorobots**

The ability of nanomotors to output external forces is the key to the practical application of various motion types. The output force of nanowire nanomotors was measured by using one gold nanowire to laterally push another gold nanowire (Fig. 4a and Supplementary Movie 10), where the force was determined to be ~5 μN (Supplementary Fig. 2). Considering of the effective absorption area (~5%), and photothermal (30%) and thermal-mechanical (0.01%) energy conversion efficiency (Supplementary Fig. 8), the transient output power of this nanomotor was ~5 μW. The corresponding thrust-to-weight ratio was as high as $10^7$, far exceeding those of any artificial machine or natural organism. In addition, the measured output force is ~0.1 μN for tilted nanowires, which is very suitable for working in tight spaces.

In some simple application scenarios, a single nanomaterial itself not only acts as a nanomotor, but also works independently as a complete nanorobot (Supplementary Fig. 7). For example, Fig. 4b shows the assembly of an artificial molecule[38] by controlling a gold nanowire to push two quantum dots (~10 nm in diameter) together, achieving the same precision as that obtained in atomic force microscopy (AFM) operations[39].

Leverage existing machine vision technology, we can further endow our nanorobots with artificial intelligence. Here, by mimicking the operation mechanisms of cleaning robots that are widely used in the macro world, we demonstrated an autonomous cleaning nanorobot working in the microscopic world (Fig. 4c, Supplementary Fig. 10, and Supplementary Movie 11). Combining technologies such as machine vision, deep learning, feedback control and path planning, this nanoplate-based nanorobot could automatically clean up nanodroplets in a target area and keep this area clean. Such cleaning nanorobots may find future applications in non-invasive cleaning of microscopic areas, such as contaminated surfaces of photosensitive microchips or optical fiber end faces.

As mentioned in the previous section, the photothermal shock mechanism is universal as changing the actuation wavelengths and target materials. Here we can drive Pd nanomaterial using a 1064-nm-wavelength pulsed laser. Due to their relatively large surfaces, nanoplate-based nanorobots have a larger payload capacity, which allows them to be equipped with more onboard components or cargos by taking existing macro-mechanical designs and thus work like the macrorobots. To demonstrate this, through a bottom-up assembly approach, we utilized various components, including Pd nanoplates (body) and Pd nanowires (tail), and cadmium selenide (CdSe) nanowires (sensor), to assemble a nanorobot with complex structures (0.6 ng in mass, Fig. 4d and Supplementary Movie 12). Because the nanorobot resembles a Chinese horseshoe crab, we name it HOUbot. Theoretically, its payload can be on the order of milligrams (million times its mass, equivalent to an ant). Here we attached a ZnO micropillar (cargo, 2.3 ng in mass) with ultraviolet (UV) adhesives. By irradiating specific body parts, basic vehicular motion could be realized (Fig. 4e). The HOUbot could be used to manipulate other nano-objects through head pushing (Fig. 4f), tail wagging (Fig. 4g) and tail stabbing (Fig. 4h). Thus, the nanorobot could realize higher degrees of freedom and perform delicate tasks. Through optical pumping, the CdSe nanowire lasers[40] could be used for in situ humidity sensing (Fig. 4i). Furthermore, our HOUbot was tested for over 10 million actuation pulse cycles with no noticeable signs of wear or performance degradation.

## Discussion

We have succeeded in developing a general actuation technique to solve nanofriction obstacle by using photothermal shock, and by using the powerful nanomotors, we further realized autonomous nanorobots working like conventional robots with precise motion control. Our nanomotors exhibit the highest thrust-to-weight ratios thus far, while achieving nanoscale step accuracy, ultralow actuation power (~μW), and nanosecond-scale response time. Just like ignition (controllable explosions) in gasoline engines, our technique utilizes a shock-induced local expansion to convert thermal energy into

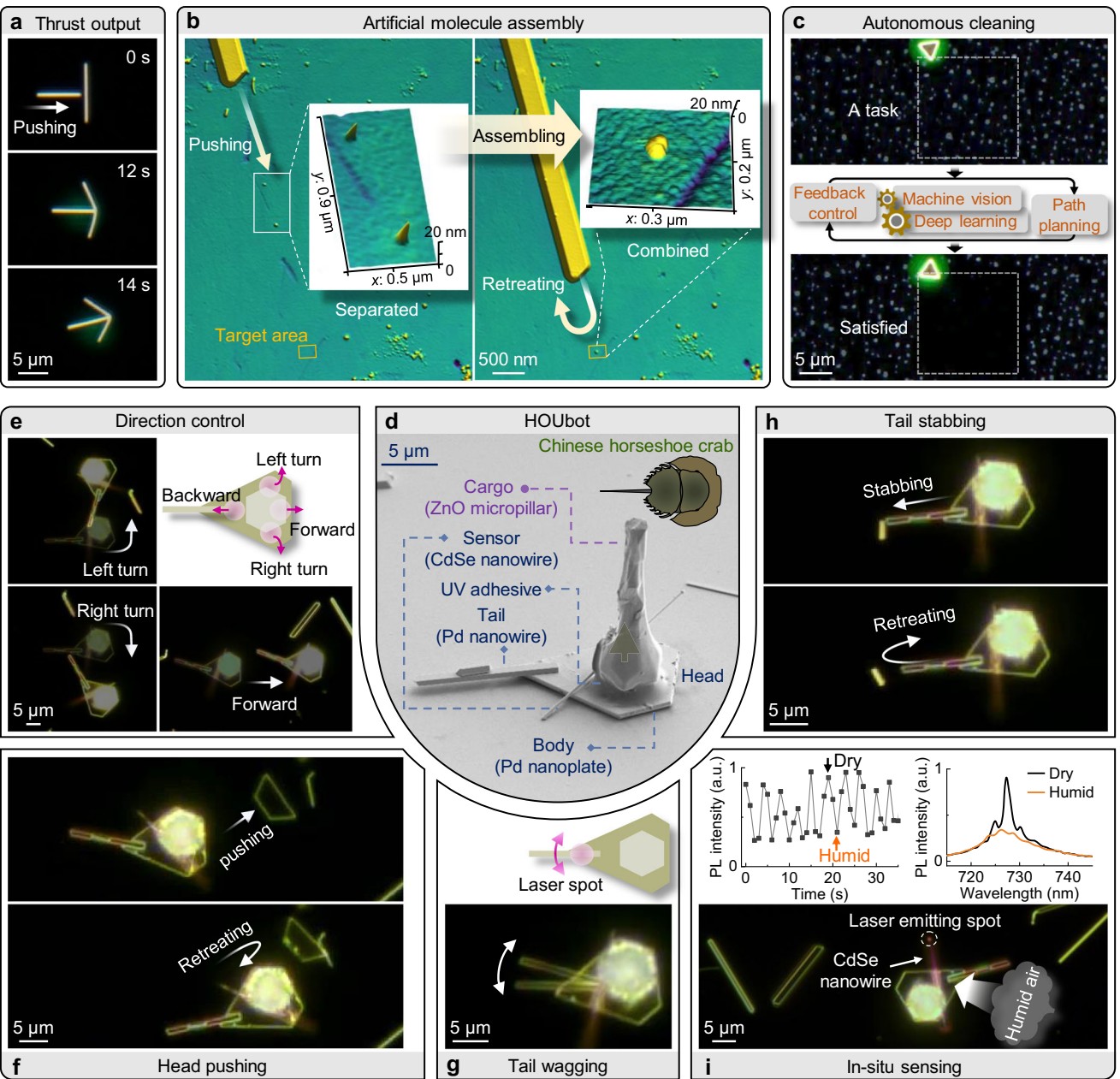

**Fig. 4 | Intelligent nanorobots with intricate motion and sophisticated functions. a** Measurement of thrust output by using one gold nanowire to laterally push another gold nanowire. **b** AFM images showing a gold nanowire pushing two separated CdSe/CdS core−shell quantum dots (from the left white box region to the right one) and combining them. Insets: close-ups of the quantum dots. **c** Autonomous cleaning nanorobot using a gold nanoplate, which could maintain the target area (top, solid box) at a satisfactory cleanliness (bottom) through machine vision, deep learning, feedback control and path planning. **d** Scanning electron microscope image of the HOUbot nanorobot, assembled by a body (Pd nanoplate), a tail (Pd nanowire), and a sensor (CdSe nanowire). The cargo (ZnO micropillar) was immobilized on the nanoplate by UV adhesives. Inset, top view schematic of a Chinese horseshoe crab. **e** Basic vehicular motion. Inset: schematic describing motion control by radiating specific body regions. Intricate motion includes head pushing (**f**), tail stabbing (**g**) and tail wagging (**h**, two superimposed photographs). **i** In situ humidity sensing by using the CdSe nanowire lasing signals. As the humidity of the local environment changed periodically, the laser peak intensity (left inset) and spectra (right inset) changed accordingly. Source data are provided as a Source data file.

mechanical energy, while nanofriction and heat dissipation play auxiliary roles in restoring the nanomotors for repeated work. This simple shock-actuation mechanism can be applied to almost any light-absorbing nanomaterials, and thus we revealed a field of transient thermoelastic dynamics at the nanoscale that can be utilized to design future nanomechanical structure.

By using the core component–nanomotors with powerful thrust to address nanotribology challenges, our work paves the way to build practical nanorobots working on dry solid surfaces. The strong thrust allows nanorobots to be equipped with complex onboard structures, such as movable end-effectors, microelectronic chips, and sensors. At the same time, with optically actuating and motion controlling, we can directly leverage existing machine vision technology, by incorporating artificial intelligence, to develop advanced autonomous nanorobots in both hardware and software. Furthermore, through spatial light modulation[24] and multi-robot collaboration, autonomous nanorobot swarms[41,42] can be realized to perform more challenging tasks and unearth unprecedented application scenarios in various fields such as nanomanufacturing, biomedicine[43,44], and aerospace in the future.

## Methods

### Materials and fabrication

Single-crystal Au and Pd nanomaterials used in our work were synthesized by the thermal evaporation method[35]. ZnO micropillars and CdSe nanowires were synthesized by the vapor–liquid–solid process[40]. We used different micro/nanomaterials as building blocks to assemble our HOUbot in Fig. 4d (main text). Pd nanoplates and ZnO nanopillars were mechanically dispersed on a silica substrate. Through manual manipulation under an optical microscope[13,35], a microdroplet of fluoropolymer was added onto the selected Pd nanoplates by a silica fiber probe. Then a ZnO micropillar was added onto a Pd nanoplate and then cured by a UV light. A CdSe nanowire and a Pd nanowire were successively added onto the Pd nanoplate.

### Photothermal-shock tweezers

Our photothermal-shock tweezers setup (Supplementary Fig. 1) is similar to traditional optical tweezers, but their work principles are completely different. A homemade nanosecond fiber laser with central wavelengths switched between 532 and 1064 nm (pulse width: ~10 ns) was used as the excitation light source. Through a collimator, free-space Gaussian-like beams with a $LP_{01}$ fundamental mode were obtained and focused upward on samples through a ×40 microscope objective with a numerical aperture (NA) of 0.6. $D_{spot}$ on the substrate could be changed from ~5 μm at the focus point to ~10 μm by a piezo $z$-stage (PIA25, Thorlabs). A piezo $xy$-stage (PIAK10, Thorlabs) was used to move the substrates to change the relative positions between the substrates and the laser spots.

A half-wave plate was used to change the polarization direction of beams. A neutral density filter was used to adjust laser intensity. Unless otherwise specified, light irradiation on the substrates and micro/nano-objects was randomly polarized. The repetition rate of the nanosecond pulse laser ranged from 100 Hz to 10 kHz. A camera was used in real-time monitoring the samples through an objective (×100, NA = 0.8). A slanted 532-nm notch filter was used to partly attenuate the spot intensity, so that samples and light spots could be observed at the same time. Compared with conventional optical tweezers, no high-NA objectives or highly-focused beams are required here.

### Autonomous control of nanorobots

The lasing trapping behavior allows for automated motion control of the nanowire via programming the paths of laser spots. Meanwhile, the camera was used for image recognition and feedback control achieved by homemade LabVIEW programs. For the autonomous nanorobot cleaning nanodroplets in Fig. 4c (main text), we used the camera to sample ~500 images of the substrate contaminated by ethanol-water mixture nanodroplets as input samples for deep neural network training, and obtained a nanodroplet model for detecting the numbers and positions of nanodroplets on the substrate through a pre-training model[45]. As a cleaning task began, we first selected a region of interest (ROI) near a cleaning nanorobot in the microscopic images as the aim area, with the camera sampling images in real-time simultaneously. The nanorobot was guided by the piezo $xy$-stage along the scanning-like planning path to suck the nanodroplets off the substrate (similar to a macro cleaning robot). As a cleaning cycle ended, the nearest neighbor frame was sampled and the nanodroplet model was used for image recognition. The nanorobot would continue cleaning cycles until no nanodroplets were detected in ROI. Thus, autonomous control of nanorobots was realized.

### In situ sensing

Semiconductor nanowires are good candidates for nanoscale lasing sources. Here the CdSe nanowire on the HOUbot was pumped by the 532-nm-wavelength nanosecond laser from top to bottom, with emissions collected by the dark-field objective. The local humidity environment around the HOUbot was changed over a period of ~4 s, and the

time-dependent lasing signals of the CdSe nanowire were collected by a spectrometer through a homemade program once per second. Thus, the time-dependent spectra and lasing peaks could be obtained, reflecting a process of in situ humidity sensing.

### Surface characterization and force measurement

We directly used commercial fused silica substrates without any polishing or other surface treatments. We selected three typical regions to determine the average surface roughness ($R_q$) via the AFM topographies (Cypher S, Oxford Instruments), with scanning square ranges of 50 nm, 1 μm, 5 μm, and 10 μm for each region (Supplementary Fig. 2a). The friction coefficient between gold micro/nano-objects and silica substrates was measured with a silica colloidal probe (CP-FM-SiO-A, NanoAndMore GmbH). The AFM probe was calibrated using the raw thermal noise spectra of the cantilever and its known spring constant[46]. By linearly fitting between measured friction and applied load, a friction coefficient of ~0.2 was obtained (Supplementary Fig. 2b).

To measure the force required to move a nanowire, we placed a probe (Multi75Al-G, Budget Sensors) at one side of a nanowire and then laterally pushed it (Supplementary Fig. 2c, inset). The applied tip load was set to 760 nN and the manipulation speed was set to 200 nm s⁻¹. The whole nanowire required to overcome the maximum static friction force is up to 5 μN (Supplementary Fig. 2d). The results also showed that the friction force is related to the area of the sliding part, and smaller contacts, such as tilted nanowires, are on the order of 0.1 μN.

### Theory analysis

The component form and summation of dummy indices are used. The commas in the subscript and the dot above the letter represent the partial differentiation and time derivative, respectively. A bold letter represents a vector (**u**), and an italic bold letter represents a tensor (**s**). Assuming small deformations, the linear elasticity theory is used. The thermoelastic coupling stress **s** needs to be subtracted from the original stress **σ** by the natural thermal expansion counterpart[47]

$$s_{ij} = \sigma_{ij} - 3K\alpha(T - T_0)\delta_{ij} = \sigma_{ij} - \beta(T - T_0)\delta_{ij}, \quad (3)$$

$$\sigma_{ij} = K\varepsilon_{ll}\delta_{ij} + 2\mu\left(\varepsilon_{ij} - \frac{1}{3}\varepsilon_{ll}\delta_{ij}\right), \quad (4)$$

$$\varepsilon_{ij} = \frac{1}{2}(u_{i,j} + u_{j,i}), \quad (5)$$

where $K = \lambda + \frac{2}{3}\mu = \frac{E}{3(1-2\nu)}$ is the bulk modulus, $\lambda = \frac{\nu E}{(1-2\nu)(1+\nu)}$ is Lamé's first parameter, $\mu = \frac{E}{2(1+\nu)}$ is Lamé's second parameter, $E$ is Young's modulus, and $\nu$ is Poisson's ratio, $\alpha$ is thermal expansion coefficient, $\beta = 3K\alpha$ is thermodynamics coupling coefficient, $\varepsilon$ is the strain, $u$ is the displacement, and $T$ is the temperature ($T_0$ is the environment temperature). The dynamic equation (with the external force **f**) of a linear elastic body is expressed by

$$s_{ji,j} + f_i - \rho\ddot{u}_i = 0. \quad (6)$$

The linear elastic dynamic equation was derived by substituting Eq. (3) into Eq. (6)

$$(\lambda + \mu)u_{j,ij} + \mu u_{i,jj} - \beta T_{,i} + f_i - \rho\ddot{u}_i = 0, \quad (7)$$

or in the vector notation,

$$(\lambda + \mu)\nabla\nabla \cdot \mathbf{u} + \mu\nabla^2\mathbf{u} - \beta\nabla T + \mathbf{f} - \rho\ddot{\mathbf{u}} = 0, \quad (8)$$

where $\nabla$ is the Nabla operator. By using the linear expansion approximation of $\nabla \cdot u \approx 3\alpha(T - T_0)$ and the condition of no external force, we derive a typical 3D wave equation for a homogeneous medium in the form of Eq. (2), with the right-hand term proportional to $\nabla T$ representing thermal gradient force ($F_{grd}$). Upon the pulsed light, $F_{grd}$ acts as an impulsive excitation and causes a series of transient vibrations, which are completely different from steady-state vibrations. It is noted that $F_{grd}$ is a volumetric force, and its volumetric integration $F_{Ngrd}$ is a force.

Considering the possible heat flux acceleration in thermal shock, Fourier's law can be modified by introducing thermal relaxation time[48]

$$q_i + \tau_0 \dot{q}_i = -kT_{,i}, \tag{9}$$

where $q_i$ represents the heat flux component (of the vector $\mathbf{q}$), $k$ is the thermal conductivity and $\tau_O$ is the relaxation time (the time lag needed to establish steady-state heat conduction). The hyperbolic-form heat conduction equation is thus obtained by

$$\rho c_p \dot{T} - kT_{,ii} + T\alpha_{ij}\dot{s}_{ij} = Q, \tag{10}$$

where $T\alpha_{ij}\dot{s}_{ij} = \alpha T\dot{s}_{ij}\delta_{ij} = \beta T(\dot{u}_{i,i} - 3\alpha\dot{T})$ and $Q$ is the heat source. The thermal conduction equation with a thermo-mechanical coupling term is given by

$$\rho c_p(\dot{T} + \tau_0 \ddot{T}) - kT_{,ii} + \alpha(\dot{s}_{ij}T + \tau_0 \ddot{s}_{ij}T + \tau_0 \dot{s}_{ij}\dot{T})\delta_{ij} = Q + \tau_0 \dot{Q}. \tag{11}$$

Due to thermo-mechanical coupling terms, Eq. (11) and the thermoelastic dynamic Eq. (7), or Eq. (8) in the vector notation, must be solved in conjunction.

Generally, vacuum light speed is given by $c = (\varepsilon_0\mu_0)^{-1/2} = \omega/k_0$, where $\varepsilon_0$ is the vacuum dielectric permittivity, $\mu_0$ is the magnetic vacuum permeability, $\omega$ is the angular frequency, and $k_0$ is the vacuum wave number. A Gaussian light is incident vertically from a dielectric ($n_d$, $z > 0$) to a metal ($z < 0$), with the beam waist $w_0$ at the interface ($z = 0$). The metal has a complex refractive index ($n_m + in_k$, $i$ is the imaginary unit), and a complex relative permittivity ($\varepsilon_m + i\varepsilon_k$). The photothermal absorption rate of metal per unit area is given by

$$\eta = \frac{2\varepsilon_k}{n_k} \frac{n_d}{(n_d + n_m)^2 + n_k^2} = \frac{4n_m n_d}{(n_d + n_m)^2 + n_k^2}. \tag{12}$$

The thermal absorption spatiotemporal distribution $Q_h$ induced by pulsed lasers in the metal is expressed by

$$Q_h(x,y,z,t) = P_0 F_k R_a R_s R_t. \tag{13}$$

Here $R_a$, $R_s$, and $R_t$ are the depth factor (m⁻¹), the surface factor (m⁻²) and the time factor (dimensionless),

$$
\begin{aligned}
R_a &= 2n_k k_0 \eta \exp(2n_k k_0 z) \\
R_s &= \frac{2}{\pi w_0^2} \exp(-\frac{2x^2 + 2y^2}{w_0^2}) \\
R_t &= \frac{1}{\sqrt{\pi}\tau R_{rep}} \exp[-\frac{(t-t_0)^2}{\tau^2}] \\
\tau &= \frac{t_p}{\sqrt{4\ln 2}}
\end{aligned}
\tag{14}
$$

where $t_0$ is the time of the pulse peak, and $t_p$ is the pulse duration. Specially, the product of $F_k$, $R_a$ and $R_s$ is the normalized heat absorption factor (m⁻³, normalized by the source power, Supplementary Fig. 3b and Fig. 2d in the main text), which represents the heat absorption power density (W m⁻³) per unit source power (W), and it can be obtained by the finite difference time domain (FDTD) simulation (by commercial software, Lumerical FDTD solutions).

The total van der Waals force can be calculated by the Lennard–Jones potential,

$$U_{L-J} = 4\xi \left[\left(\frac{\sigma}{r}\right)^{12} - \left(\frac{\sigma}{r}\right)^6\right], \tag{15}$$

where $r$ is the distance between two particles, $\xi$ is the depth of the potential at the minimum, and $\sigma$ is the distance at which $U_{L-J}$ is zero. We make a simplification that the distance between is never closer than the direct contact distance $r_0$, so $U_{L-J}$ becomes a simplified potential $U_{sp}$:[49]

$$U_{sp} = \begin{cases} -\frac{C}{r^6}, & r \geq r_0 \\ \infty, & r < r_0 \end{cases}, \tag{16}$$

where $C = \sigma^6$ is the London coefficient in the particle–particle pair interaction[50]. The interaction potential $U_{P-S}$ between a particle and a semi-infinite solid is derived by the volumetric integration for the solid region of the $U_{sp}$,

$$U_{P-S} = \int_V U_{sp} n dV = -\frac{\pi Cn}{6h^3}, \tag{17}$$

where $n$ is the number density of the solid, and $h$ is the distance between the particle and the solid S. Attraction per unit volume is derived by the differential at the distance of $h$

$$f_V = -\frac{d(n' U_{P-S})}{dh} = \frac{\pi Cnn'}{2h^4} = \frac{A}{2\pi h^4}, \tag{18}$$

where $A = \pi^2 Cnn'$ is the Hamaker constant between two solids. The Hamaker constants between the solid 1 and 2 can be given by[51]:

$$A_{12} = \pi^2 Cn_1 n_2 = \sqrt{A_1 \cdot A_2}, \tag{19}$$

where $A_1 = \pi^2 Cn_1^2$, $A_2 = \pi^2 Cn_2^2$ are the Hamaker constants of the solids 1 and 2, respectively, and $n_1$, $n_2$ are the number density of the solids 1 and 2, respectively. The Hamaker constant between gold and silica is estimated to be $1.88 \times 10^{-19}$ J, and those of gold and silica are $4.0 \times 10^{-19}$ J and $8.86 \times 10^{-19}$ J, respectively[52,53]. The adhesion force per unit area with the gap $d_0$ is given by the thickness integration of Eq. (18)

$$\sigma_S = \int_{d_0}^{\infty} f_V dh = \frac{A}{6\pi d_0^3}, \tag{20}$$

which is also commonly used to estimate the van der Waals attraction between two infinite parallel plates.

### Simulation setup
The FDTD method was used to solve the thermal coupling, while the finite element method (FEM, by commercial software, COMSOL) was used to solve the problems of transient thermoelastic dynamic. To get closer to the real adhesion force, we use the form of body force in Eq. (18), which decays rapidly at the boundary, rather than the surface force in Eq. (20). The effects of both localized surface plasmon resonances and boundaries are contained in a linear factor $F_k$. Heat absorption is assumed to only occur on the light-facing side of the nanowire in the FEM simulation. The penalty method and the Coulomb friction law are used for estimating contact pressure and friction. The gap between the nanowire and the silica substrate is set to be 0.5 nm, with the average roughness values of the nanowire 0.23 nm and that of the substrate 0.27 nm (Supplementary Fig. 2a). The friction coefficient is assumed to be 0.2 (Supplementary Fig. 2b). Values of parameters used in the simulation are listed in Supplementary Table 1.

## Data availability

Numerical data underlying the figures presented in this study are provided in the Source data file. All data are available upon request from the corresponding author. Source data are provided with this paper.

## Code availability

All code used for the work is available upon request from the corresponding author. The code used in autonomous control of nanorobots are publicly available in the Zenodo public repository at https://doi.org/10.5281/zenodo.10060042, ref. 54.

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

## Acknowledgements

We thank Professor Wei Fang at Zhejiang University for helpful suggestions. F.G. thanks the funding from the National Natural Science Foundation of China (grant nos. 62122054 and 62075131) and the Natural Science Foundation of Shanghai (21ZR1481100).

## Author contributions

F.G. conceived the idea and supervised this project. Z.G. performed data analysis and simulations. R.Z. performed the most experiments. T.S. performed the material preparation and force characterizations. H.L. assisted with the automated control testing. F.G., Z.G., R.Z., and T.S. co-wrote the paper. L.D., Y.L., X.L., J.L., and S.Z. provided many insightful suggestions.

## Competing interests

The authors declare no competing interests.
