## [Peer Review File · Nature Communications]

Reviewers' Comments:

Reviewer #1:

Remarks to the Author:

The authors have investigated a propulsion method for nanowire manipulation that exploits thermal shock to overcome nanoscale friction. While the concept appears to be novel, some clarifications are needed regarding the proposed propulsion mechanism. The authors explained how thermal shock overcomes friction in the case of point-end contact mode, but they did not sufficiently explain the mechanisms involved in line contact and shape bending. Additionally, authors should provide a clear explanation of the term "thermal gradient force." To the reviewer's understanding, thermal gradient force refers to the force induced by the temperature gradient established along the nanowire. However, this term could be confused with thermophoresis.

Furthermore, is there a possibility that the optothermal trapping force and convection (or sudden thermal expansion) explain the mechanism?

Regarding bending, it is necessary to provide a thorough explanation related to solid mechanics, including forces, stress-strain relationships, and other relevant factors.

With regard to the seven-nanowire cluster, it is unclear how the wires can be separated independently. Further clarification is needed regarding the size, beam, and resolution.

Reviewer #2:

Remarks to the Author:

The authors demonstrate a nanomotor that can move on dry surfaces (solid-gas interface) by photothermal shock. A photothermal-shock tweezer was developed to overcome the nanofriction and actuate metallic nanowires to perform controlled motions. Utilizing the thrust produced by photothermal shock, the nanorobots can be assembled with complex onboard structures, and can be applied in different scenarios. The results are interesting, and the content is well-organized. Here are some concerns:

- (1) The light-driven motion of metallic nanowires on the silica surface has been previously demonstrated by the authors (Nat. Commun., 2021,12, 385). What are the new findings and advancements of this work?
- (2) Interactions between nanoparticles and substrates should be introduced in the Introduction section. Besides mechanical forces from the nanowire deformation, the authors should consider other possible mechanisms that may contribute to the propulsion of the gold nanowires.
- (3) The surface properties of the substrate are also expected to play a vital role for the propulsion of the nanowire motors. The authors should investigate the light-driven motions of the nanowire motors on different substrates, such as polymer, metallic, and silicon substrates, and discuss the substrate effect.
- (4) Since the contact end is free and can move with the light spot, why can the nanowire keep at a tilted state rather than lying on the substrate during propulsion?
- (5) What are the processes of light-induced photothermal deformation of metallic nanowires? Please explain this mechanism briefly.
- (6) The authors should discuss the influence of the size and shape of the light spot on the motions of the gold nanowires.
- (7) Does the pulse frequency influence the motion of the line-contact motor?
- (8) To further verify the photothermal-shock mechanism, control experiments should be conducted by using those nanoparticles without the photothermal effect.
- (9) The authors demonstrated the motions of point-contact and line-contact nanomotors using gold nanowires. Why do they choose Pd instead of Au nanowires when assembling the nanorobot?
- (10) As the nanomotors move with the light spot, they can not be considered to be "autonomous". There are also some typos in the manuscript (e.g., line 2, p2 and Supplementary Figure 2d).
- (11) This metallic nanowire motor is not the first-ever one that can move on dry surfaces (Adv. Mater., 2015, 27, 3883; Nat. Commun., 2015, 6, 7310; Nat. Commun., 2021,12, 385; PNAS, 2023, 120, e2221740120). I also found some references related to light-driven motors (Chem.

Soc. Rev. 2017, 46, 6905; Adv. Mater. 2017, 29, 1603374. Natl. Sci. Rev. 2021, 8, nwab066; Research 2022, 2022, 9816562).

Reviewer #3:

Remarks to the Author:

The article introduces a novel nanorobot propulsion system on dry surfaces that utilizes a nanosecond pulse laser. This innovative method effectively overcomes the friction challenges posed by dry substrates, in contrast to traditional optical tweezers. Through both experimental and simulation approaches, the authors reveal the propulsion mechanism of the nanorobot, which involves the local expansion of nanomaterials triggered by the laser pulse, leading to the conversion of thermal energy into mechanical energy.

The results are clearly presented, and the manuscript is well-written and easy to read. However, it would be beneficial to further clarify the driving mechanism discussed in the article. The article may be published in Nature Communications after the authors have considered the following points.

1. Considering that the instantaneous power of every laser pulse is much larger than the average power, is it reasonable to fully neglect the effect of the momentum of photons compared to the driving force caused by the deformation of robots?
2. Building on the previous point, authors should consider comparing the gradient forces caused by the light power and thermal gradient forces. Some experimental proof is needed to distinguish the pulsed light vs cw light.
3. As each pulse the particle will move a certain range, the speed should be increased significantly with repeating higher frequency, can author provide that?
4. How does friction come into play during the expansion and contraction of the nanowire that lies on the substrate?
5. The author may consider using L^2/α (L : length of nanowire, α : thermal diffusivity in gold) to evaluate the time scale of thermal conduction in the nanowire.
6. If increasing the laser power on the tilted nanowire, will the wire fly off the substrate?
7. In line 22 on Page 5, the sentence "Since F_{int} is proportional to the light intensity, the nanowire leaves from the light spot center along its tilt direction" is confusing. Authors might consider rephrasing it.
8. The unit of gold density in Supplementary Table 1 is not correct.

Response Letter to *Nature Communications* (Manuscript No.: NCOMMS-23-22642)

Title: "Autonomous nanorobots with powerful thrust under dry solid-contact conditions by thermal photoshock"

Author(s): Zhaoqi Gu, Runlin Zhu, Tianci Shen, Lin Dou, Hongjiang Liu, Yifei Liu, Xu Liu, Jia Liu, Songlin Zhuang, Fuxing Gu

Response to Reviewer's comments:

Reviewer #1

The authors have investigated a propulsion method for nanowire manipulation that exploits thermal shock to overcome nanoscale friction. While the concept appears to be novel, some clarifications are needed regarding the proposed propulsion mechanism.

The authors explained how thermal shock overcomes friction in the case of point-end contact mode, but they did not sufficiently explain the mechanisms involved in line contact and shape bending. Additionally, authors should provide a clear explanation of the term "thermal gradient force." To the reviewer's understanding, thermal gradient force refers to the force induced by the temperature gradient established along the nanowire. However, this term could be confused with thermophoresis. Furthermore, is there a possibility that the optothermal trapping force and convection (or sudden thermal expansion) explain the mechanism? Regarding bending, it is necessary to provide a thorough explanation related to solid mechanics, including forces, stress-strain relationships, and other relevant factors. With regard to the seven-nanowire cluster, it is unclear how the wires can be separated independently. Further clarification is needed regarding the size, beam, and resolution.

Comment (1): *The authors explained how thermal shock overcomes friction in the case of point-end contact mode, but they did not sufficiently explain the mechanisms involved in line contact and shape bending.*

Reply to Comment (1): Thank you very much for the positive response to our work. We fully agree with this suggestion, and it is very helpful for improving our work. To address this issue in detail, we reorganized the mechanism, which of both line contact and shape bending can be described by three processes as following:

- (i) How the nanowire responds under a single pulse;
- (ii) How the nanowire responds under repeated pulses, as the spot remaining stationary;
- (iii) How the nanowire responds as the light spot moves.

The mechanism of line contact:

(i) Responding under a single pulse: As the nanowire is initially shocked, its hotter end can overcome the friction and slide, while the other end (colder end) with more contact area has larger friction, and it has not been affected by the thermal elastic disturbance, so the colder end remains stationary. This unbalanced displacement is responsible for the net centroid displacement of the nanowires. Except for this initial non-uniform displacement, the nanowire expands or contracts as a whole, hardly affecting the net centroid displacement.

(ii) Responding under repeated pulses and the stationary spot: The nanowire will step repeatedly under repeated pulses. As the nanowire is penetrating into the light spot, the advancing end (hotter end) will cross the spot center into the region with the opposite light intensity gradient, and this end will be subjected to the thermal gradient force, which is opposite the advancing direction and will become larger and larger. As the centers of the nanowire and the light spot roughly coincide, the opposite two thermal gradient forces on the nanowire balance each other, and thus the nanowire will no longer step forward. (The phenomenon that the nanowires automatically move toward the stationary spot center is similar to the trapping behavior of conventional optical tweezers, although the mechanism is completely different.)

(iii) Responding as the light spot moves: When the light spot moves, the nanowire will repeat the Step (ii) to catch up with the light spot to be trapped, so the nanowire motion can be completely controlled by that of the light spot.

The mechanism of bending shape:

(i) Responding under a single pulse: When the centers of the nanowire and the light spot coincide, the thermal gradient forces still exist, and they squeeze the nanowire towards the center at both ends respectively. Then, the nanowire is stable in the axis direction, but unstable in the lateral direction. If the nanowire and the spot are not strictly coincident, or the nanowire itself has certain initial lateral bending, the nanowire will deviate from the equilibrium and rapidly bend. Each pulse intensifies the bending tendency, which is typical positive feedback.

(ii) Responding under repeated pulses and the stationary spot: Repeated pulses will finally bend the nanowire axis and move the nanowire into the position of an iso-intensity line of the light spot. Since the thermal gradient force is related to the light intensity gradient, as long as the nanowire axis crosses the light intensity line, it will be squeezed by the thermal gradient force. The nanowire will finally stop because each part absorption of the nanowire after bending is uniform, the temperature difference is very small, and the thermal gradient force was too small to overcome the friction.

(iii) Responding as the light spot moves: When the spot moves, the nanowire axis recrosses with the iso-intensity line, so Step (ii) is repeated. The reason why the nanowire can remain curved is the interface friction, rather than the lattice destruction or melting.

In the revised manuscript, we have modified related mechanisms in the main text.

Comment (2): *Authors should provide a clear explanation of the term "thermal gradient force." To the reviewer's understanding, thermal gradient force refers to the force induced by the temperature gradient established along the nanowire. However, this term could be confused with thermophoresis.*

Reply to Comment (2): We thank the reviewer for this comment. Let's start with the derivation of the formula. Deriving the equation of thermoelastic coupling,

$$(\lambda + \mu)\nabla\nabla \cdot \mathbf{u} + \mu\nabla^2\mathbf{u} - \beta\nabla T + \mathbf{f} - \rho\ddot{\mathbf{u}} = 0,$$

using the linear thermal expansion approximation of $\nabla \cdot \mathbf{u} \approx 3\alpha(T - T_0)$, eliminating the term of $\nabla\nabla \cdot \mathbf{u}$, arranging the equation into a brief forced-vibration form,

$$\rho\ddot{\mathbf{u}} - \mu\nabla^2\mathbf{u} \approx \mu\alpha\nabla T.$$

The right-hand term is named after **thermal gradient force** (F_{grd} , volumetric force), for a temperature gradient in its expression. There are two terms on the left hand of the equation: the first term is the dynamic term and the second term is the stress term (representing the stress distribution). Force can not only cause deformation (stress term) but also motion (dynamic term). At the beginning of thermal shock, the stress loading of various parts of the nanowire is not in place in time, so F_{grd} can cause instantaneous motion (the mechanism of line contact and bending shape). The change of stress after thermal shock will be stable, which will cause stable vibrations characterized by geometric size. These vibrations distributed throughout the nanowire are generally called elastic waves, but obviously, the forces caused by elastic waves is not as strong as that of transient thermal shock. This can be likened to the general forced vibration, which usually starts with anharmonic transient vibration.

There is a fundamental difference between thermal gradient force and thermophoresis force. The thermal gradient force occurs in an independent solid body, the source of this force is the thermal stress inside the body, with the help of friction to realize overall displacement. It can be explained by solid mechanics (or thermoelasticity). The temperature gradient occurs on the object itself. Thermophoresis is thermal diffusion related phenomenon, a statistical description of collective particle motion, and tends to form a macroscopic, distinguishable flow that drives migration of a particle group. Temperature gradients occur throughout the environment medium.

In the revised manuscript, we modified related discussions in the introduction of the main text.

Comment (3): *Is there a possibility that the optothermal trapping force and convection (or sudden thermal expansion) explain the mechanism?*

Reply to Comment (3): As mentioned in the comment (2), our driving force source is completely the stress inside the solid, without the involvement of environmental media particles, so these environmental media-related factors (the optothermal trapping force and convection) are unlikely to be the driving mechanism.

There is also a very important factor, that is, the magnitude order of the force. For the nanowires we studied, the magnitude order of the adhesion force and friction force

are μN , while those of forces based on fluids or traditional optical tweezers are only pN. These forces are not enough to overcome the friction force to drive the nano-objects.

In addition, the expression "sudden" in parentheses has certain physical rationality. Here, we borrow the existing concept of thermal shock to describe something that happens suddenly and rapidly, and extend it to the well-defined "photothermal shock" in this work. The driving mechanism consists of the rapid expansion of centroid displacement forward during shock and the slow contraction of centroid displacement almost stationary after shock. Therefore, it cannot be simply expressed by "sudden thermal expansion", while the shock process includes both initial expansion and later contraction.

Comment (4): *Regarding bending, it is necessary to provide a thorough explanation related to solid mechanics, including forces, stress-strain relationships, and other relevant factors.*

Reply to Comment (4): We thank the reviewer for this comment. The key point in the bending process is that the pulsed laser gives a positive feedback on the bending. The initial small lateral shift of gold nanowires can increase rapidly under the pulse, for which we provide simulated evidence that a 6- μm -length gold nanowire of an initial bend of 10 degrees further laterally shifts as the pulsed light illuminated at its center. We also provide time-dependent simulation results of velocity, acceleration, strain, and strain rate, in which the data is taken at the geometric center of the nanowire. The simulation results are given as following:

In the revised manuscript, the related discussions have been integrated into the bending-shape part of comment (1).

Comment (5): *With regard to the seven-nanowire cluster, it is unclear how the wires can be separated independently. Further clarification is needed regarding the size, beam, and resolution.*

Reply to Comment (5): We thank the reviewer for this comment. The specific process of the separation of seven nanowires is shown in Supplementary Video 5. The light spot

is not specially treated, with a repetition rate of 100 Hz, an average power of 0.7 μ W, and a spot radius of 4.3 μ m.

From the perspective of force analysis, the nanowires stuck together are subject to friction from the substrate as well as from other nanowires (since we are concerned with tangential forces, we only discuss friction). For a nanowire, if the friction between it and the other nanowires is greater than the friction between it and the substrate, then it and the other nanowires as a whole will follow the light spot, which can be observed in early period of Supplementary Video 5. Since the shock force of the nanowire is in all directions, the relative positions of the nanowires can be disrupted by photothermal shock, or the adhesion forces between different nanowires can be weakened. The friction between the nanowires varies with time, while the friction between the nanowire and the substrate does not change much. Once the friction between a nanowire and the other nanowire is less than the friction between it and the substrate, it will be separated from the nanowire cluster.

The size of the spot affects the optical power density or light intensity, and the absorption cross section of the nanowire is limited, so the intensity must be ensured to be strong enough, otherwise the light spot cannot drive the nanowire.

The step resolution is on the magnitude of nanometer, and it changes with the relative position between the nanowire and the light spot. Figure 2c shows the time-dependent experimental displacements of the nanowire as it was trapped in the light spot.

In the revised manuscript, we have added the parameter of the used laser (the spot radius of 4.3 μ m) in Supplementary Video 5.

Reviewer #2

The authors demonstrate a nanomotor that can move on dry surfaces (solid-gas interface) by photothermal shock. A photothermal-shock tweezer was developed to overcome the nanofriction and actuate metallic nanowires to perform controlled motions. Utilizing the thrust produced by photothermal shock, the nanorobots can be assembled with complex onboard structures, and can be applied in different scenarios. The results are interesting, and the content is well-organized. Here are some concerns:

Comment (1): *The light-driven motion of metallic nanowires on the silica surface has been previously demonstrated by the authors (Nat. Commun., 2021,12, 385). What are the new findings and advancements of this work?*

Reply to Comment (1): Before we address individual comments, we first would like to thank the reviewer for the positive response to our work. Your comments are very insightful and are very helpful to improve our manuscript.

In this work, we have demonstrated arbitrarily complex motion control of nanowires on a two-dimensional plane, consisting of basic lateral and axial motions.

We have extended the drive mechanism to nanoplates, and further realized nanorobotic applications. Our previous work (*Nat. Commun.*, 2021, 12, 385) only realized going forward and backward on the microfibers, without the high-degree-of-freedom motion control and the trapping behaviour, and our explanation for driving mechanism was not clear. The originally introduced surface wave or elastic wave has now been shown to be only a by-product of the shock effect, not the fundamental source of driving at all. Although we found that friction played an important role, we did not notice the shock effect. Of course, friction is not the fundamental source, but it is still an important auxiliary factor in driving nanowires.

The elastic wave disperses the mechanical energy to the whole nanowire, so the thrust induced by elastic wave is much smaller than that of the initial shock. Due to the lack of continuous excitation (compared to pulse excitation), the elastic wave after thermal shock will rapidly decay under the damping of interface friction. Therefore, we think that the driving factor of elastic wave is negligible compared to that of the initial shock. In fact, we have discussed elastic waves in the original manuscript.

Comment (2): *Interactions between nanoparticles and substrates should be introduced in the Introduction section. Besides mechanical forces from the nanowire deformation, the authors should consider other possible mechanisms that may contribute to the propulsion of the gold nanowires.*

Reply to Comment (2): We thank the reviewer for this comment. For the interaction between the target object and the substrate, we mainly consider the van der Waals force. Electrostatic forces can be excluded because objects are in contact with each other, and it is difficult to accumulate charges on the metal, and there is no applied electric field. Capillary forces can be excluded because we operate at a dry solid interface, with low humidity and no liquid film between interfaces.

Optical force, thermophoresis force, photophoresis force, optothermal trapping force and other force induced by convection (mentioned by the reviewer #1) have been proposed to driving particles in fluid environments, but here we excluded these forces, because the magnitude order of these forces (\sim pN) is obviously too small to driving subjects (nanowires and nanoplates) we investigate. In our origin manuscript, we have excluded the effect of elastic waves, and we have discussed it in the comment (1).

In addition to the thermal shock, the ablation and melting will also occur under high energy pulsed laser. Ablation and melting destroy the material itself. If ablation occurred, we would be able to observe the ejection of material onto the substrate, and the material would rapidly decrease due to loss under continuous pulse action, but here our materials can be used for a long time without significant damage, which were confirmed by AFM characterization. Similarly, the factor of melting phase transition can also be excluded.

In the revised manuscript, we have added some related discussions about the interaction force and other driving forces in the introduction of the main text.

Comment (3): The surface properties of the substrate are also expected to play a vital role for the propulsion of the nanowire motors. The authors should investigate the light-

driven motions of the nanowire motors on different substrates, such as polymer, metallic, and silicon substrates, and discuss the substrate effect.

Reply to Comment (3): It is a very insightful comment. We used a beam upward on samples through a microscope objective, so we need to choose transparent substrates, such as silica, polystyrene, MgF₂ and silica covered by monolayer MoS₂. All substrates above could be used to achieve driving nanowires, and no mechanical scratches were observed among these substrates except polystyrene.

Monolayer materials, such as monolayer MoS₂, with only a few layers of atoms, tend to be fragile and prone to damage. However, in our experiments, the nanowires completely slide across the monolayer MoS₂-covered silica substrate, and no significant difference between the monolayer MoS₂ was observed by AFM characterization. This is one of the reasons why we declare the driving method of photothermal shock is wear-free.

After driving the nanowires on a polystyrene substrate, a narrow channel was found on the substrate, which is not mechanical damage, but thermal damage, because of the relatively low melting temperature of polystyrene (~200 K). According to the simulation results, although the maximum instantaneous temperature of the nanowires (~400 K in Fig. 1d and ~600 K in Supplementary Fig. S5) is much lower than the melting point of the material itself (1337 K), it is sufficient for polystyrene to melt. In the revised manuscript, we have added these results as a new figure (and the caption) in the Supplementary Figure 9 and related discussions in the main text.

Caption:

Different substrates for the photothermal shock driving. (a) Sequential superimposed photographs of a gold nanowire sliding across a monolayer MoS₂ on a silica substrate. (Bottom) Comparison of high-resolution AFM images of surface changes on the monolayer before and after nanowire sliding.

Optical (b) and AFM (c) images of PS substrates, and enlarged AFM 3D images (d). visible scratches were observed under the optical microscope on the polystyrene substrate, and the AFM surface characterization revealed that the axis of the nanowires delineated a channel with a width of ~ 80 nm and a depth of ~ 30 nm. The channel on the polystyrene substrate comes from thermal damage.

Comment (4): *Since the contact end is free and can move with the light spot, why can the nanowire keep at a tilted state rather than lying on the substrate during propulsion?*

Reply to Comment (4): The nanowires synthesized by a chemical vapor deposition method have good monocrystalline properties, and their surfaces grow along specific crystal faces, so the end shape of the nanowires are half-octahedron, rather than arc or other shapes. Therefore, the nanowires can stand stably against one of the crystal faces. It is interesting that the angle between tilted nanowire axis and the substrate is usually 60 degrees. A scanning electron microscopy image of a Pd nanowire is provided as following. The crystal forms of Au and Pd are face-centered cubic, so the shape of Au nanowires is similar.

Comment (5): *What are the processes of light-induced photothermal deformation of metallic nanowires? Please explain this mechanism briefly.*

Reply to Comment (5): Since we used nanosecond light source, ultrafine hot electron processes can be ignored (applying to time scale of femtoseconds or picoseconds), otherwise a two-temperature model can be used.

According to our simplified model, the linear expansion approximation is adopted, that is $\nabla \cdot \mathbf{u} \approx 3\alpha(T - T_0)$, the volume expansion ratio of the nanowire is completely controlled by the temperature difference. But this is only local behavior, the overall thermoelastic behavior needs to be determined by the following formula,

$$\rho \ddot{\mathbf{u}} - \mu \nabla^2 \mathbf{u} \approx \mu \alpha \nabla T .$$

The excitation of thermal gradient force can cause both motion and deformation.

Thermal expansion can be divided into non-uniform expansion caused by temperature gradient and uniform expansion caused by simple temperature rising. The former will cause the centroid displacement of the nanowire, which occurs at the initial stage of thermal shock, while the latter does not cause centroid displacement and continues occurring for a long period of time after thermal shock.

Therefore, the photothermal deformation of nanowires can also be divided into photothermal shock deformation and ordinary non-shock photothermal deformation. The former corresponds to the non-uniform expansion occurring in the initial phase of photothermal shock and can cause the centroid displacement. The latter corresponds to uniform expansion after thermal shock.

Comment (6): *The authors should discuss the influence of the size and shape of the light spot on the motions of the gold nanowires*

Reply to Comment (6): We thank the reviewer for this comment. The size of the spot affects the optical power density or light intensity, and the absorption cross section of the nanowire is limited, so the intensity must be ensured to be strong enough, otherwise the light spot cannot drive the nanowire. Similar research results are shown in Supplementary Figure. 6, instead of changing the spot size, we use the covering area (covering length) to study the driving of nanowires.

We have also utilized the spatial light modulation to realize the manipulation by annular light spots. The nanowire and nanoplate was trapped in the annular sector, rather than the dark center. This suggests that the key factor in manipulating nanowires is the light intensity gradient.

In the revised manuscript, we have added the experimental movie of the spatial light modulation to Supplementary Movie 8 and related discussions in the main text.

Comment (7): *Does the pulse frequency influence the motion of the line-contact motor?*

Reply to Comment (7): Fixing the single pulse shape (the single pulse duration and energy), if a single pulse cannot drive the nanowire, then no matter how to increase the pulse repetition rate (even melting), pulsed laser will not drive the nanowire. If a single pulse can drive the nanowire and increase the repetition rate, it is true that the motion speed of the nanowire can be significantly increased, but it is not a linear increase.

Our basic assumption now is that all the effects that can be caused by the current pulse will disappear before the next pulse, if the repetition rate is too high, the pulse interval is too short, and each pulse will not be independent. We also need to point out that the nanowire motion is controlled by the light spot, so as the nanowire is guided by the light spot, the speed of the nanowire speed is entirely determined by that of the light (the speed we give manually or programmed), not by the pulse repetition rate, unless the light spot moves so fast that the nanowire cannot catch up.

The comment (3) of reviewer #3 also refer to this issue. In the revised manuscript, we have added related discussions in the main text.

Comment (8): *To further verify the photothermal-shock mechanism, control experiments should be conducted by using those nanoparticles without the photothermal effect.*

Reply to Comment (8): It is a very insightful comment. Silica is a commonly used material without the photothermal effect in the wave band we used. We then used silica balls (4 μm and 8 μm in diameters), silica nanowires and silica nanoplates, and we neither observed them being driven or damaged (melted). In the revised manuscript, we have discussed this issue in the comment (1) of reviewer #3, and added related discussion into Supplementary Note 1.

Comment (9): *The authors demonstrated the motions of point-contact and line-contact nanomotors using gold nanowires. Why do they choose Pd instead of Au nanowires when assembling the nanorobot?*

Reply to Comment (9): The reason that we choose Pd material is mainly to improve the thermal shock resistance. There are four key parameters (melting point, thermal diffusivity, Young's modulus and Thermal expansion coefficient), and Thermal diffusivity measures the rate of transfer of heat of a material from the hot end to the cold end, so it is more accurate than thermal conductivity in heat transfer analysis. Pd material has three favorable parameters (higher melting point, higher Young's modulus, lower thermal expansion coefficient) and one unfavorable parameter (lower thermal diffusivity), because there are more favorable parameters, the thermal shock resistance of Pd is higher, which has been proven by experiments that Pd material has higher damage power thresholds. We have also achieved driving silver nanowires. The parameter table of three materials (Au, Pd and Ag) is given as following, in which the arrows after their term names indicate the trends favorable to thermal shock resistance.

	Au	Pd	Ag
Melting point (K) \uparrow	1337.33	1828.05	1234.93
Thermal diffusivity ($\text{mm}^2\cdot\text{s}^{-1}$) \uparrow	127.68	24.46	174.02
Density ($\text{g}\cdot\text{cm}^{-3}$)	19.30	12.02	10.49
Thermal conductivity ($\text{W}\cdot\text{m}^{-1}\cdot\text{K}^{-1}$)	318.00	71.80	429.00
Specific heat capacity ($\text{J}\cdot\text{Kg}^{-1}\cdot\text{K}^{-1}$)	129.05	244.13	235.01
Young's modulus (GPa) \downarrow	79.00	121.00	83.00
Thermal expansion coefficient (10^{-6}K^{-1}) \downarrow	14.20	11.80	18.90

In fact, the wavelength we used has also changed from 532 nm to 1064 nm, in order to keep away from the absorption wave band of the sensor (CdSe) of HOUbot. The absorption ratio of Au is weaker at 1064 nm, while that of Pd is not. This actually proves the generality of photothermal shock: It is not limited to a specific material, a specific wavelength, or a specific shape.

In the revised manuscript, we have added the experimental movies of Ag into Supplementary Movie 8 and related discussions in the main text.

Comment (10): *As the nanomotors move with the light spot, they can not be considered to be “autonomous”. There are also some typos in the manuscript (e.g., line 2, p2 and Supplementary Figure 2d).*

Reply to Comment (10): We thank the reviewer for this comment. From the perspective of conventional robots, motion control is the core factor in robotic system. Autonomous robots are intelligent machines capable of performing tasks in the real world by themselves, without explicit human control [Ref.1]. In our work, the motion control of the nanorobots is achieved by combining with artificial intelligence technology, without any manual intervention, so it can be considered autonomous. The motion control of an autonomous robot can be loaded inside, or assisted by the external environment, and the cleaning nanorobot we have realized is the latter. Our HOUbot, on top of the cleaning task, is capable of higher dimensional and more detailed movement of internal components, as well as in-situ sensing.

The word “autonomous” may refer to another type of autonomous motors, whose propulsions stem from chemical gradients of catalytic generation [Refs.2 and 3]. In the case, the autonomous motion means the nanomotors move through self-electrophoresis or self-propulsion by creating their own local fields, without any external magnetic, electric or optical field to guide their motions.

Therefore, we think that our nanorobots on dry surfaces, working like conventional robots with precise motion control, is indeed autonomous. We have added the words “motion control” and “conventional robots” in the revised manuscript to clarify this issue. We have also corrected grammatical and spelling errors in the related places where the reviewer mentioned and throughout the article.

- [1] Bekey, G. A. *Autonomous robots: from biological inspiration to implementation and control.* (MIT Press, 2005).
- [2] Paxton, W. F. et al. Catalytic nanomotors: autonomous movement of striped nanorods. *J. Am. Chem. Soc.* **126**, 13424–13431 (2004).
- [3] Wang, W., Castro, L. A., Hoyos, M. & Mallouk, T. E. Autonomous motion of metallic microrods propelled by ultrasound. *ACS Nano* **6**, 6122–6132 (2012)

Comment (11): *This metallic nanowire motor is not the first-ever one that can move on dry surfaces (Adv. Mater., 2015, 27, 3883; Nat. Commun., 2015, 6, 7310; Nat. Commun., 2021,12, 385; PNAS, 2023, 120, e2221740120). I also found some references related to light-driven motors (Chem. Soc. Rev. 2017, 46, 6905; Adv. Mater. 2017, 29, 1603374. Natl. Sci. Rev. 2021, 8, nwab066; Research 2022, 2022, 9816562).*

Reply to Comment (11): Thank you for pointing out this issue that may have caused misunderstanding. In fact, the word “first-ever” is used to describe our nanorobots, and we actually also agree with that we are not the first nanomotors on the solid interface from the perspective of the nanomotor driving. However, judging from the clarification of the word “autonomous” in the previous comment (10), our nanorobots are indeed the first to realize autonomous motion control on a solid interface.

Almost nanomotors in previous reports only achieving actuating, without further motion control. For motors, it is sufficient to convert other energies into mechanical energy and complete actuation. However, for robots that sense, think and act, only the completion of actuation is far insufficient, and the more important destination to be solved is the motion control after actuation, and further the ability to perform tasks in complex environments. In this work, we completed destinations from actuation to motion control, and further nanorobot applications.

We are also grateful to the reviewers for providing many valuable references, and we have cited some typical references in the revised manuscript.

Reviewer #3

The article introduces a novel nanorobot propulsion system on dry surfaces that utilizes a nanosecond pulse laser. This innovative method effectively overcomes the friction challenges posed by dry substrates, in contrast to traditional optical tweezers. Through both experimental and simulation approaches, the authors reveal the propulsion mechanism of the nanorobot, which involves the local expansion of nanomaterials triggered by the laser pulse, leading to the conversion of thermal energy into mechanical energy.

The results are clearly presented, and the manuscript is well-written and easy to read. However, it would be beneficial to further clarify the driving mechanism discussed in the article. The article may be published in Nature Communications after the authors have considered the following points.

Comment (1): *Considering that the instantaneous power of every laser pulse is much larger than the average power, is it reasonable to fully neglect the effect of the momentum of photons compared to the driving force caused by the deformation of robots?*

Reply to comment (1): Thank you very much for the highly positive response to our work. Your comments are very insightful and are very helpful to improve our manuscript. We also notice the potential effects of photon momentum in physics, but it is not the driving mechanism in this work, for which we give four reasons as following:

(i) We will first give numerical simulation evidences. With a typical beam waist radius of 4.3 μm , an average power of 1 μW , and repetition rate of 100 Hz, (instantaneous peak power 1 W, instantaneous light intensity $I = 1.7 \text{ MW/m}^2$), we use the Lorentz formula to calculate the instantaneous optical force of a 10- μm -length gold nanowire and the net force is $1.7 \times 10^{-10} \text{ N}$, that is, 0.17 pN. This value is obviously much smaller than the μN force of friction that the nanowire is subjected to.

(ii) From the instantaneous power conversion of light momentum action, we can also consider the light pressure P , estimated with $P = 2rI/c$, where I is the light intensity,

r is the reflectivity, and c is the speed of light. Let $r = 1$ (full reflection) to get the maximum light pressure. According to the simulated data $I = 1.7 \text{ MW/m}^2$, the optical pressure is calculated to be 5.6 mPa, while the Young's modulus of Au is 75 Gpa, so the strain is lower than 10^{-14} , which is equivalent to no deformation.

(iii) We can estimate the light gradient force by the following formula,

$$F_{\text{ogrd}} = \frac{2\pi n_0 a^3}{c} \left(\frac{m^2 - 1}{m^2 + 2} \right) \nabla I,$$

where n_0 is the index of the target object, a is the particle radius (characteristic size) and m is the relative index between the object and background. We set $n_0 = 0.54$ (real part of index for Au in 532 nm), $m = 0.54$, $a = 1 \text{ }\mu\text{m}$, and $I = 1.7 \text{ MW/m}^2$. In Gaussian spot, the maximum light gradient occurs at half the radius of the beam waist. The maximum light gradient force is 10^{-15} N , (if the characteristic size increases correspondingly, it may be increased to the magnitude order of pN). Many previous experiments have proven that for micro/nano-objects, the optical force provides forces of pN, while the adhesion force and friction force are in the magnitude order of μN . The pN force is sufficient to drive particles in fluid environments, but obviously not suitable in a solid interface. Strong adhesion and friction are the difficulties of solid interface driving.

In the previous three reasons, numerical simulation results were used to confirm that **the light force is in the pN order**. The thermal gradient force confirmed to be on the order of μN in the manuscript. We also provide experimental evidences for the next reason.

(iv) Non-absorbable materials, without the photothermal effect, such as micrometers silica balls, cannot be driven, which we have discussed in the comment (8) of reviewer #2. In this case, the light force, or the force generated by the photon momentum transfer, still exists, but does not work in driving. This excludes the possibility of pure optical driving.

In fact, we have discussed momentum transfer from photons in the original manuscript. In the revised manuscript, we have also added some related discussions about optical forces in the introduction of the main text.

In order to clarify this important issue, we have separately added this relevant discussions in this comment as new Supplementary Note 1.

Comment (2): *Building on the previous point, authors should consider comparing the gradient forces caused by the light power and thermal gradient forces. Some experimental proof is needed to distinguish the pulsed light vs cw light.*

Reply to comment (2): We thank the reviewer for this comment. The comparison between optical gradient force and thermal gradient force has been complete in reason (iii) in comment (1). CW light cannot drive the nanowires, even if the laser power is high enough to make the nanowires melt. We give the snapshots of experimental movies as following:

Comment (3): *As each pulse the particle will move a certain range, the speed should be increased significantly with repeating higher frequency, can author provide that?*

Reply to comment (3): It is true that the faster the repetition rate of the pulsed laser, the faster the motion of the nanowire. We have used an 8- μm -length gold nanowire under the pulsed laser of different repetition rate and observed that the trapping process in the laser spot is obviously fast under higher repetition rate. We give the snapshots of experimental movies as following. Comment (7) of reviewer #2 has also refer to a similar issue. In the revised manuscript, we have added related discussions in the main text.

Comment (4): *How does friction come into play during the expansion and contraction of the nanowire that lies on the substrate?*

Reply to comment (4): It is a very insightful comment. In common sense, friction hinders movement, but in practical applications, almost all ground vehicles rely on friction driving.

Friction is not a fundamental force. In solid mechanics analysis, friction is described by the frictional stress at the contact boundary (numerically, derived from the surface integral of the frictional stress at the contact boundary). In the original manuscript, we only discussed $F_{N\text{pro}}$ and $F_{N\text{grd}}$ in equation (2), which are both volumetric integration of the volumetric force, but bound by the boundary condition (the friction condition). The distributions of boundary forces and the volumetric forces are two sides of the same issue. Therefore, we can describe the shock driving mechanism either by the volumetric forces ($F_{N\text{pro}}$ and $F_{N\text{grd}}$), or by the boundary condition (friction). In Supplementary Figure 5, the net force in (b) is numerically equal to the entire friction force in (e).

When lying nanowire experiences a photothermal shock, the thermal gradient force can only make its neighboring region (hotter end) to overcome friction and slide

in its direction. The remaining (colder end) is governed by static friction and remains relatively stationary with the substrate. So far, the driving function of the single pulse has been basically completed, and the remaining thing is only heat transfer and dissipation, making the nanowires to the initial state to meet the next pulse.

Friction plays a very important auxiliary role. Friction in the colder end counteracts the initial shock of the thermal shock, ensuring that only hotter end is sliding and the colder end is stationary, so that the center of mass of the entire nanowires will move. The friction force is non-uniform at the beginning. From the time-dependent simulation results of displacement and friction (Supplementary Figure 5e), it can be seen that the change of friction at the colder end is later than that at the hotter end.

When the colder end begins to slide, the thermal shock begins to weaken and the centroid displacement of the nanowire begins to slow down. After the thermal shock, the nanowire begins to evolve to thermal equilibrium, when the nanowire centroid displacement will slightly decrease, but is ignored compared with the huge advancing displacement during the thermal shock.

The friction force ensures the nanowires to step forward greatly during the thermal shock, and prevents the centroid displacement after the thermal shock to return to the origin, so the net centroid displacement of the nanowires under a single pulse is achieved.

In the revised manuscript, we have added the related discussions of friction in the line-contact driving mechanism of the main text.

Comment (5): *The author may consider using L^2/α (L : length of nanowire, α : thermal diffusivity in gold) to evaluate the time scale of thermal conduction in the nanowire?*

Reply to comment (5): Thanks for the reviewer's suggestion, and the use of heat transfer characteristic time is indeed a good method. The thermal diffusivity (α) is defined as following

$$\alpha = \frac{k}{\rho c_p},$$

where k is the thermal conductivity, ρ is the density and c_p is the specific heat capacity. α is about 128 mm²/s for gold. The typical nanowire length is 6~10 μ m, and the heat transfer characteristic time is 0.5~0.8 μ s. Therefore, the thermal shock driving time is almost consistent with the pulse duration, about 10 ns, which is much less than the heat transfer characteristic time. We use the concept of shock to show that the shock effect is much faster than the slow thermal process. That is, the thermal shock drive has been completed, but the thermal equilibrium has not been completed.

Comment (6): *If increasing the laser power on the tilted nanowire, will the wire fly off the substrate?*

Reply to comment (6): Very insightful comment! Indeed, they do fly off, including tilted nanowires and lying nanowires. The adhesion between nanowire and the substrate can be weakened or destroyed by the pulsed laser. We

have provided an new experimental movie that nanowires bounce off the substrates in Supplementary Movie 6 in the revised manuscript.

Comment (7): *In line 22 on Page 5, the sentence “Since F_{int} is proportional to the light intensity, the nanowire leaves from the light spot center along its tilt direction” is confusing. Authors might consider rephrasing it.*

Reply to comment (7): The original statement in our manuscript replied that objects always tend to have lower energies, but a more physical explanation is lacking. Thanks to the reviewer to point it out, we will switch to a more physical term.

Tilted nanowires are driven by the principle that the thermal inertia force provides the stepping force and the thermal gradient force provides the direction control force.

When a tilted nanowire is in the light spot center, the thermal inertia force is large, but the thermal gradient force is small (the central light intensity is large but the gradient is small), so the nanowires move almost in a straight line and the direction change is small, which is similar to centrifugation.

When the nanowire is away from the center of the spot, the thermal inertia force decreases and the thermal gradient force correspondingly increases, so the swing angle of the nanowire in a single pulse also increases. In analogy with the circular motion, it can be considered that the linear velocity of the nanowires decreases, while the centripetal acceleration increases. When the single-pulse step and angle swing of the nanowire meet the conditions of circular motion, the nanowires can move approximately around the light spot.

In the revised manuscript, we have modified the related discussions in the main text.

Comment (8): *The unit of gold density in Supplementary Table 1 is not correct.*

Reply to comment (8): We thank the reviewer to point out such an elementary error. We have corrected it in the revised manuscript.

Reviewers' Comments:

Reviewer #1:

Remarks to the Author:

The authors have successfully addressed all comments. I recommend acceptance.

Reviewer #2:

Remarks to the Author:

All of my concerns have been appropriately addressed by the authors, and I believe the manuscript is now suitable for acceptance in its current version.

Reviewer #3:

Remarks to the Author:

The revised paper has addressed most my questions and I agree that it can be published on Nature Commun. However, please recheck the light instantaneous light intensity, which should be $1.7 \times 10^{10} \text{ W/m}^2$, rather than $1.7 \times 10^6 \text{ W/m}^2$ (MW/m²).

Response to referees to *Nature Communications* (Manuscript

No.: NCOMMS-23-22642A)

Title: "Autonomous nanorobots with powerful thrust under dry solid-contact conditions by thermal photoshock"

Author(s): Zhaoqi Gu, Runlin Zhu, Tianci Shen, Lin Dou, Hongjiang Liu, Yifei Liu, Xu Liu, Jia Liu, Songlin Zhuang, Fuxing Gu

Response to Reviewer's comments:

Reviewer #1

The authors have successfully addressed all comments. I recommend acceptance.

Reply to Comment: Thank you very much for the positive response to our work.

Reviewer #2

All of my concerns have been appropriately addressed by the authors, and I believe the manuscript is now suitable for acceptance in its current version.

Reply to Comment: Thank you very much for the positive response to our work.

Reviewer #3

The revised paper has addressed most my questions and I agree that it can be published on Nature Commun. However, please recheck the light instantaneous light intensity, which should be 1.7×10^{10} W/m², rather than 1.7×10^6 W/m² (MW/m²).

Reply to Comment: Thank you very much for the positive response to our work and pointing out the error. In the revised manuscript, we have changed related values in Supplementary Note 1.